# A novel gene *REPTOR2* activates the autophagic degradation of wing disc in pea aphid

Erliang Yuan[1,2], Huijuan Guo[1,2], Weiyao Chen[1,2], Bingru Du[3], Yingjie Mi[1], Zhaorui Qi[3], Yiyang Yuan[4], Keyan Zhu-Salzman[5], Feng Ge[4]*, Yucheng Sun[1,2]*

[1]State Key Laboratory of Integrated Management of Pest Insects and Rodents, Institute of Zoology, Chinese Academy of Sciences, Beijing, China; [2]CAS Center for Excellence in Biotic Interactions, University of Chinese Academy of Science, Beijing, China; [3]School of Life Science, Hebei University, Baoding, China; [4]Institute of Plant Protection, Shandong Academy of Agriculture Sciences, Jinan, China; [5]Department of Entomology, Texas A&M University, College Station, United States

*For correspondence:
gef@ioz.ac.cn (FG);
sunyc@ioz.ac.cn (YS)

Competing interest: The authors declare that no competing interests exist.

**Abstract** Wing dimorphism in insects is an evolutionarily adaptive trait to maximize insect fitness under various environments, by which the population could be balanced between dispersing and reproduction. Most studies concern the regulatory mechanisms underlying the stimulation of wing morph in aphids, but relatively little research addresses the molecular basis of wing loss. Here, we found that, while developing normally in winged-destined pea aphids, the wing disc in wingless-destined aphids degenerated 30-hr postbirth and that this degeneration was due to autophagy rather than apoptosis. Activation of autophagy in first instar nymphs reduced the proportion of winged aphids, and suppression of autophagy increased the proportion. *REPTOR2*, associated with TOR signaling pathway, was identified by RNA-seq as a differentially expressed gene between the two morphs with higher expression in the thorax of wingless-destined aphids. Further genetic analysis indicated that *REPTOR2* could be a novel gene derived from a gene duplication event that occurred exclusively in pea aphids on autosome A1 but translocated to the sex chromosome. Knockdown of *REPTOR2* reduced autophagy in the wing disc and increased the proportion of winged aphids. In agreement with REPTOR's canonical negative regulatory role of TOR on autophagy, winged-destined aphids had higher *TOR* expression in the wing disc. Suppression of *TOR* activated autophagy of the wing disc and decreased the proportion of winged aphids, and vice versa. Co-suppression of *TOR* and *REPTOR2* showed that ds*REPTOR2* could mask the positive effect of ds*TOR* on autophagy, suggesting that *REPTOR2* acted as a key regulator downstream of *TOR* in the signaling pathway. These results revealed that the TOR signaling pathway suppressed autophagic degradation of the wing disc in pea aphids by negatively regulating the expression of *REPTOR2*.

## Editor's evaluation

This study addresses the genetic basis of an iconic example of developmental plasticity. The background and hypothesis are clear, and the experiments are well conducted. The paper shows that winglessness in aphids involves autophagy (rather than apoptosis) of the wing primordium, and that this is regulated by a novel gene, REPTOR2, which is a target of TOR signaling.

## Introduction

Flight ability is a great evolutionary success for animals that favors seeking new habitats, mates, and food resources (*Roff, 1990*). The loss of such ability has occurred in numerous insects (*Roff, 1990*; *Harrison, 1980*; *Roff and Fairbairn, 1991*). The wingless morph has evolved to optimize resource allocation from dispersal to reproduction, which facilitates population expansion under suitable conditions (*Zera and Denno, 1997*; *Mole and Zera, 1993*). Alternation between the wing morphs in aphid offspring is strongly influenced by maternal experience (*Braendle et al., 2006*). Pea aphids are recognized as a compelling laboratory model to study regulatory mechanisms underlying wing dimorphism, most notably through maternal population density (herein refers to as maternal density). It has been shown that ecdysteroids, miR-9b-ABCG4-insulin signaling, and two laterally transferred viral genes (*Apns-1* and *Apns-2*) are responsible for producing winged offspring in parthenogenetic females (*Vellichirammal et al., 2017*; *Shang et al., 2020*; *Parker and Brisson, 2019*). Nevertheless, the phylogenetic evidence suggests that winged morph in aphids is a default developmental pattern, while the wingless morph is secondarily derived, from their winged ancestor (*Braendle et al., 2006*; *Miyazaki, 1987*). Thus, how wingless morph is controlled under favorable conditions needs to be studied, as such information will help to elucidate the differentiation of wing dimorphism in aphids.

Tissue degradation is a common developmental process in insects executed by programmed cell death (PCD), such as autophagy and apoptosis (*Xu et al., 2019*). During metamorphosis, *Drosophila* salivary glands, midgut and other larval tissues are removed by PCD (*Xu et al., 2019*). For example, suppression of apoptosis or autophagy or both could delay the degeneration of salivary glands (*Berry and Baehrecke, 2007*). Suppressing autophagy, but not apoptosis, severely interferes with the removal of the midgut during metamorphosis (*Denton et al., 2012*). These findings suggest that tissue-specific degeneration required different types of PCD. Furthermore, histological examinations in pea aphids show that the second instar nymphs of the winged-destined morph obviously possess a pair of wing discs, whereas wingless-destined morph does not (*Ishikawa et al., 2008*; *Ogawa and Miura, 2013*). Since the wing bud and the full wing are developed from the wing disc, it would be interesting to know whether and how PCD controls the fate of the wing disc cells, ultimately resulting in the wingless morph.

The evolutionarily conserved nutrient-sensing TOR (the target of rapamycin) pathway has been implicated in regulation of fecundity, life span as well as tissues degradation of insects under different nutritional conditions (*Xu et al., 2019*; *Katewa and Kapahi, 2011*). TOR coordinately regulates the balance between growth and autophagy in response to cellular and physiological conditions as well as environmental stresses such as intracellular amino acid availability and accessible energy levels (*Saxton and Sabatini, 2017*). Shortage of dietary amino acid depletion or starvation, inhibits TOR activity, which is known as a crucial step for autophagy induction in eukaryotes (*Shimobayashi and Hall, 2016*). Downregulation of TOR signaling pathway seems to be necessary for autophagic degradation to occur in different insect tissues such as the salivary gland and midgut (*Berry and Baehrecke, 2007*; *Denton et al., 2012*). TOR suppresses autophagy by directly phosphorylating multiple residues of autophagy-related proteins or repressing the transcription of autophagy-related genes. TOR phosphorylates Atg13, causing reduced affinity between Atg1 and its binding proteins and subsequently repressed autophagy (*Puente et al., 2016*). Alternatively, TOR also phosphorylates Ser527 and Ser530 of REPTOR (repressed by TOR) to prevent its entry into nucleus where REPTOR targets autophagy-related genes and suppresses their expression (*Tiebe et al., 2015*). In this study, we identify a novel gene *REPTOR2* in pea aphids by comparing the transcriptome of wingless-destined versus winged-destined aphids at 24-hr postbirth. Since *REPTOR2* has a higher expression in thorax of wingless-destined aphids, we hypothesized that *REPTOR2* activates autophagic degradation in the wing disc to produce the wingless morph. Likewise, winged-destined aphids have a stronger TOR activity to suppress autophagy in the wing disc to develop into full wings. Experimentally, we specifically determined: (i) the time point to initiate PCD in the wing disc, (ii) the PCD type that was responsible for wing disc degeneration, and (iii) the role of REPTOR2 and TOR signaling cascade in determination wing dimorphism.

## Results

### Effects of maternal density and duration on the proportion of winged offspring

To efficiently sample winged offspring, we examined how maternal density and duration affected the proportion of winged offspring. Compared to the adults individually housed, the adults housed in groups of 2, 4, and 8 in a petri dish for 8-hr led to higher proportions of winged offspring, respectively (*Figure 1A*). Since two adults were sufficient to produce over 80% winged offspring, we then tested different contact time periods on the proportion of winged morph (*Figure 1B*). Apparently, 4-hr was sufficient to induce >90% winged offspring, and 2-hr induced a moderate effect (approximately 50%) (*Figure 1B*). Furthermore, two adults with 4-hr contact duration continuously produced over 80% winged offspring for 3 days, followed by a small reduction afterward (*Figure 1C*). Furthermore, to eliminate a strain-specific induction, three other pea aphid strains were also tested for two adults and 4-hr duration; even the strain with the lowest proportion of winged offspring exceeded 40% (*Figure 1—figure supplement 1*).

### Developmental differentiation in the wing and wing disc of winged versus wingless morphs

Previous studies indicated that the wing disc was degenerated at the second instar in wingless morph of the pea aphid (*Ishikawa et al., 2008*; *Ogawa and Miura, 2013*). We compared the wing morphology across different nymphal stages between the two wing morphs. Winged- and wingless-destined aphids could not be distinguished morphogenetically until the third instar (*Figure 1D*). The wing bud of winged-destined aphids became visible during the third and fourth instars, which later developed into full unfolded wing once adult emergence. By contrast, wingless-destined aphids did not possess obvious wing buds at the third and fourth instar stages. To further determine the time point of initiation of the wing disc degeneration, the first instar nymphs at 24-, 30-, and 36-hr postbirth were collected for histological analyses. The wing disc sizes were not significantly different between the two morphs 24-hr postbirth, but were then rapidly degenerated in wingless-destined morphs while continuously developed and expended in winged-destined morphs 30-hr postbirth (*Figure 1E and F*).

### Autophagic rather than apoptotic degradation was responsible for degeneration of wing disc

Since wing disc degeneration initiated at 30-hr post birth, PCD occurred in wing disc at this time point was characterized for its type and process by transmission electron microscopy (TEM), immunofluorescence, TdT-mediated dUTP Nick-End Labeling (TUNEL), and quantitative polymerase chain reaction (qPCR). Typical autophagic features such as numerous cytoplasmic vacuoles, large autophagosomes with sequestered visible remnants of organelles, and membranous whorls were observed by TEM within the cells of the wing disc of the wingless-destined morph. By contrast, only mitophagy and a few cytoplasmic vacuoles were found in wing disc cells of winged-destined morph. Signs of autophagy were observed in the wing disc of 54% cells of the wingless-destined morph, but only 15% cells of winged-destined aphids (*Figure 2A*). Furthermore, cell shrinkage, plasma membrane blebbing, apoptotic body, chromatin condensation, nucleolus disorganization, and nuclear fragmentation, typical features of apoptosis, were hardly observed in wing discs of either winged- or wingless-destined morphs.

The immunofluorescence and TUNEL results showed that autophagy, instead of apoptosis, was activated in wing disc cells of the wingless-destined morph (*Figure 2B and C*). The autophagy-related genes *ATG1a*, *ATG1b*, *ATG3a*, *ATG3b*, *ATG4*, *ATG10*, *ATG13*, and *ATG14* had higher expression in the wingless-destined morph (*Figure 2D*), while all apoptosis-related genes except for *Decay2* were not significantly different between two wing morphs (*Figure 2E*). These results indicated that autophagy rather than apoptosis was responsible for degeneration of the wing disc in pea aphids, and thereby reduced the proportion of the winged morph. To verify this hypothesis, the group with two adults placed in a petri dish for 2-hr was applied to produce a moderate proportion (approximately 50%) of winged offspring (*Figure 2—figure supplement 1*). Newborn nymphs were fed with autophagy agonist rapamycin or antagonist 3-MA for 24-hr. As anticipated, rapamycin reduced winged

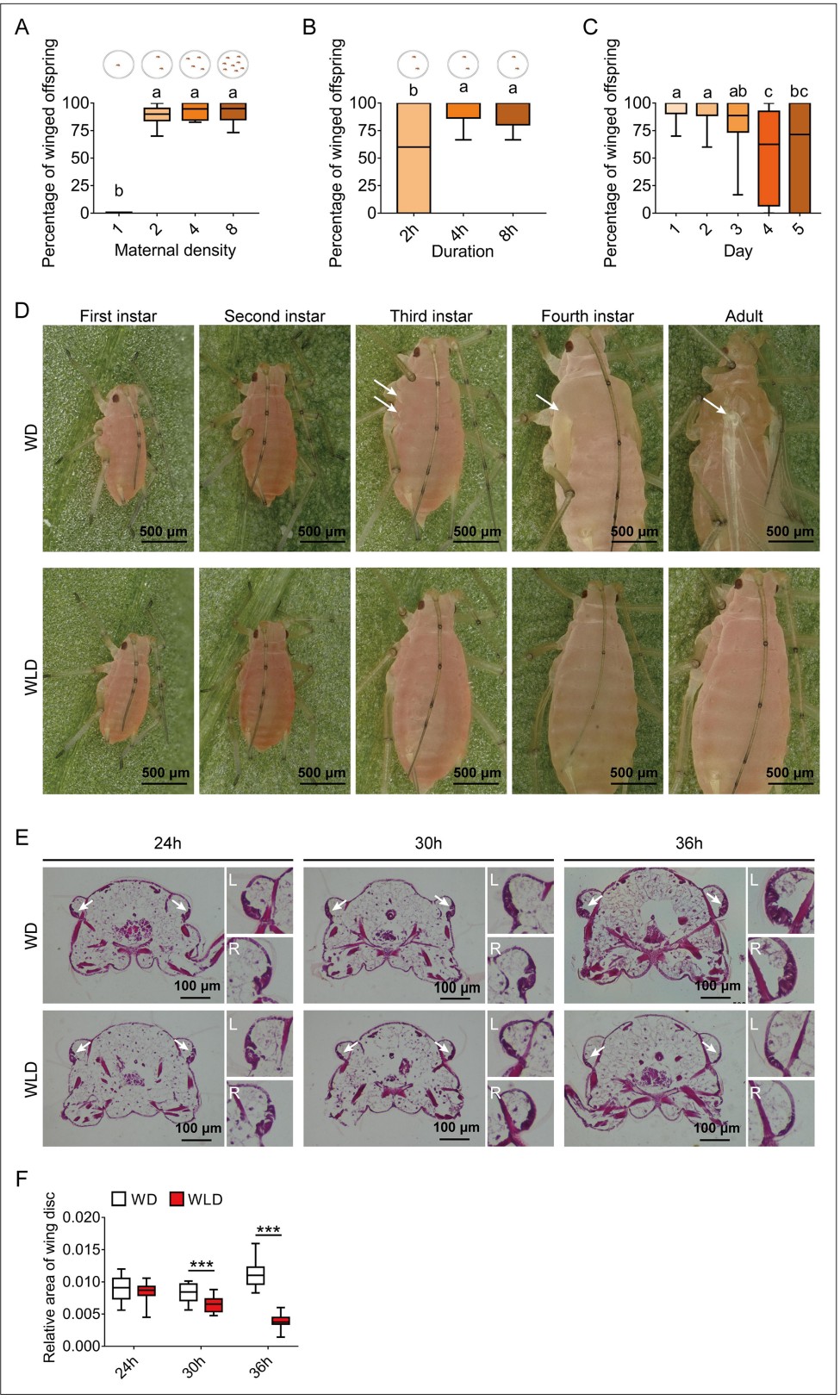

**Figure 1.** Wing dimorphism in pea aphids is transgenerational with high sensitivity to maternal density and duration, and first instar nymphs at 30-hr postbirth is a critical stage for developmental plasticity of the wing disc. (**A**) Two female adults in a petri dish can efficiently induce a high proportion of winged offspring. (**B**) Percentage of winged offspring produced by the two-adult contacting treatment for different durations. Circles in the

*Figure 1 continued on next page*

*Figure 1 continued*

diagrams above the figure panels represent petri dishes. (**C**) Daily percentage of winged offspring for the two-adult contacting treatment for 4-hr. (**D**) Developmental morphology of aphid wings from the first instar nymphal to adult stages in winged and wingless aphids. White arrows: locations of the wing bud or wing in the winged morph. (**E**) Histological comparisons of wing disc in the first instar nymphs at 24-, 30-, and 36-hr postbirth, and wing discs existed in both winged- and wingless-destined aphids. White arrows indicate the wing disc. L, left wing disc; R, right wing disc. (**F**) The relative area of the wing disc measured by ImageJ. WD, winged-destined; WLD, wingless-destined. Boxes show the interquartile range, and the line is the median value of each group (n>16). Tukey's multiple range tests at p<0.05 were used to compare means, and different lowercase letters indicate significant differences. Independent sample t test was used to compare means of the relative area of the wing disc, andsignificant difference at p<0.001 is indicated by asterisks (***).

The online version of this article includes the following source data and figure supplement(s) for figure 1:

**Source data 1.** Related to *Figure 1A*.

**Source data 2.** Related to *Figure 1B*.

**Source data 3.** Related to *Figure 1C*.

**Source data 4.** Related to *Figure 1F*.

**Figure supplement 1.** The percentage of winged offspring produced by two-adult contacting treatment for 4-hr of different pea aphid strains.

**Figure supplement 1—source data 1.** Related to *Figure 1—figure supplement 1*.

proportion from 67% to 36% by enhancing autophagy in the wing disc of first instar nymphs (*Figure 2F and H*), whereas 3-MA attenuated autophagy in the wing disc and increased the winged aphids from 54% to 67% (*Figure 2G and I*).

## A novel gene *REPTOR2* in pea aphid positively regulated the wing disc autophagy

Of the 387 differentially expressed genes (DEGs), in total, the first instar nymphs of the wingless-destined morph had 220 genes expressed higher and 167 genes expressed lower than the winged-destined morph at 24-hr postbirth (*Figure 3A*). Strong evidence has suggested that autophagic cell death is regulated by TORC1 (*Berry and Baehrecke, 2007*; *Denton et al., 2012*), we therefore focused on genes in the TOR signaling pathway (*Figure 3B*) that had significantly higher expression in wingless-destined morph than the winged counterpart. Only *REPTOR2* (*repress by TOR 2*, LOC103309809) and *SLC38A9* (*sodium-coupled neutral amino acid transporter 9*, LOC100158916) fell into this category. *SLC38A9* is a lysosomal membrane-resident protein involved in the amino acid transport and responsible for amino acids-induced TORC1 activation (*Rebsamen et al., 2015*). *REPTOR* has been recognized as a transcription factor that relays the TOR signaling cascade by activating the expression of autophagy-related genes (*Tiebe et al., 2015*).

When aligned with the *Acyrthosiphon pisum* genome, two REPTOR homologs, *ApREPTOR1* (LOC100168197) and *ApREPTOR2* (LOC103309809), were identified. Seven other aphid species contained only a single REPTOR copy, including *Myzus cerasi*, *Myzus persicae*, *Melanaphis secchari*, *Aphis gossyppii*, *Rhopalosiphum padi*, *Rhopalosiphum maidis*, and *Eriosoma lanigerum*. Both *REPTOR* genes in *A. pisum* consisted of a conserved basic region leucine zipper (BRLZ) domain (*Figure 3C*). An unrooted phylogenetic tree with bootstrap consensus was constructed across eight aphid species using coding sequence alignments (*Figure 3D*). MUMmer analyses were used to determine the similarity and homologous regions of *ApREPTOR2* in four aphid species with chromosome-level genomes in Aphidbase. *ApREPTOR1* and *MpREPTOR* had 9 exons, while *ApREPTOR2* had 7 exons (*Figure 3E*). *ApREPTOR1* shared high sequence similarity with *MpREPTOR*, *RmREPTOR*, and *ApREPTOR2*, but not with *ElREPTOR*. The variations in the flanking region of REPTOR indicated that *ApREPTOR2* was quite different from *ApREPTOR1* (*Figure 3F*). Chromosomal distribution results further showed that *REPTOR1* was located in A1 chromosome while *REPTOR2* was located in the X chromosome of *A. pisum* (*Figure 3G*). These results indicated that *REPTOR1* in pea aphid may have experienced a gene duplication to generate a novel gene *REPTOR2* which probably translocates from autosome A1 to the sex chromosome.

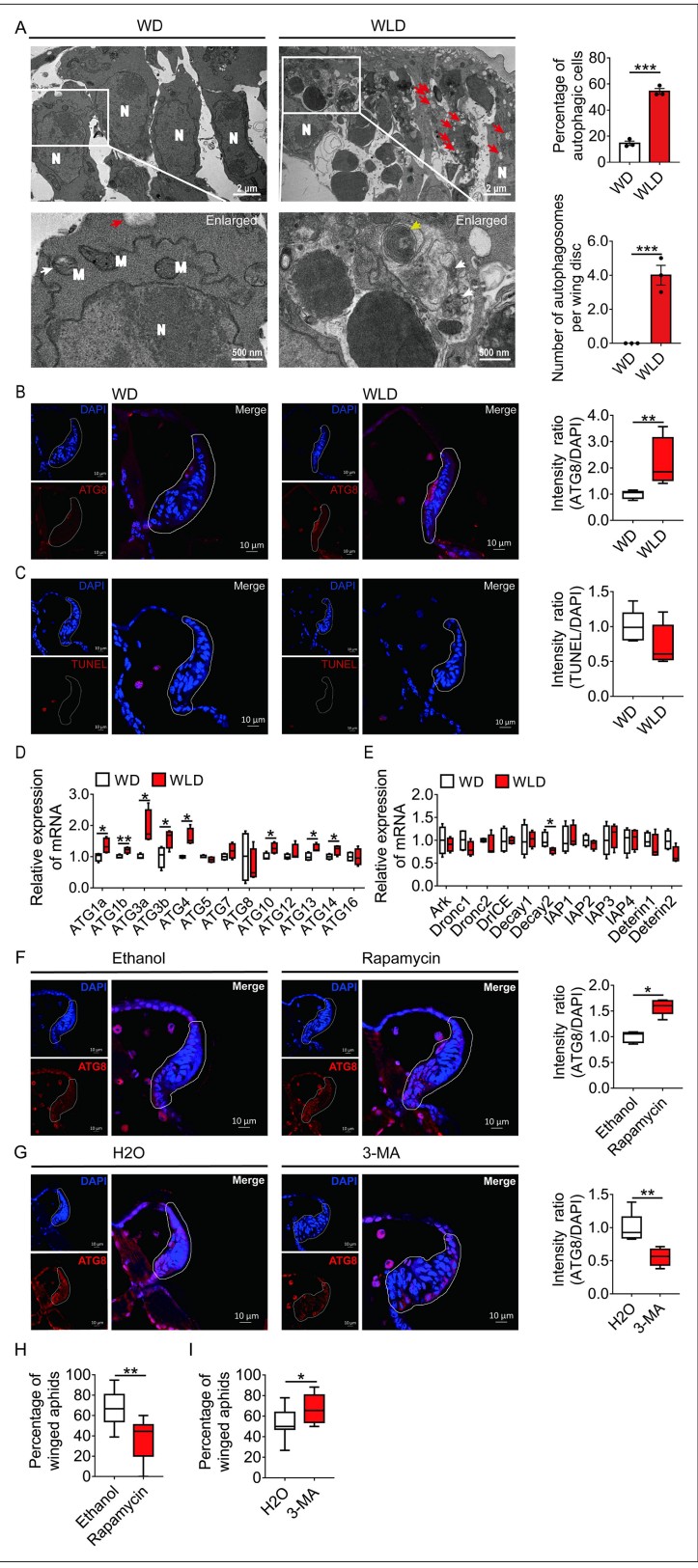

**Figure 2.** Autophagy, and not apoptosis, is responsible for degeneration of the wing disc in wingless-destined aphids. (**A**) TEM images of wing discs of winged- and wingless-destined aphids at 30-hr postbirth. N, nucleus; M, mitochondrion. Yellow and white arrows indicate autophagosomes containing membranous whorls and remnants of cellular organelles, respectively. Red arrows indicate the vacuoles (n=3). Values in bar plots are means (± SEM).

*Figure 2 continued on next page*

*Figure 2 continued*

(**B**) Autophagy in the wing disc was indicated by the hallmark ATG8 (red) in immunofluorescence. The nuclei (blue) were stained by DAPI (n=7). (**C**) Apoptosis was determined by TUNEL assays (red) (n=5). (**D**) Autophagy-related genes (*ATG1a, ATG1b, ATG3a, ATG3b, ATG4, ATG5, ATG7, ATG8, ATG10, ATG12, ATG13, ATG14,* and *ATG16*) were determined by qPCR (n=4). (**E**) Pro-apoptotic genes (*Ark, Dronc1, Dronc2, DrICE, Deterin1,* and *Deterin2*), anti-apoptotic genes (*IAP1, IAP2, IAP3, IAP4, Decay1,* and *Decay2*) were determined by qPCR (n=4). (**F**) The effect of autophagy agonist rapamycin, and (**G**) autophagy inhibitor 3-MA on autophagy in the wing disc. Autophagy was noted by ATG8 (red) in immunofluorescence, and nuclei were stained by DAPI (blue). All relative intensity was quantified by ImageJ (n>5). (**H**) The effect of autophagy agonist rapamycin, and (**I**) autophagy inhibitor 3-MA on the proportion of winged aphids. Ethanol and H$_2$O were used as control, respectively (n>9). Wing disc shown in confocal was highlighted by white circle. WD, winged-destined; WLD, wingless-destined. Boxes show the interquartile range, and the line is the median value of each group. Independent sample t test was used to compare means, and significant differences between treatments are indicated by asterisks, *p<0.05, **p<0.01, ***p<0.001. qPCR, quantitative polymerase chain reaction; TEM, transmission electron microscopy.

The online version of this article includes the following source data and figure supplement(s) for figure 2:

**Source data 1.** Related to *Figure 2A*.

**Source data 2.** Related to *Figure 2B*.

**Source data 3.** Related to *Figure 2C*.

**Source data 4.** Related to *Figure 2D*.

**Source data 5.** Related to *Figure 2E*.

**Source data 6.** Related to *Figure 2F*.

**Source data 7.** Related to *Figure 2G*.

**Source data 8.** Related to *Figure 2H*.

**Source data 9.** Related to *Figure 2I*.

**Figure supplement 1.** Experimental design.

---

*REPTOR2* had higher expression in the wingless-destined morph at 12- and 24-hr postbirth than that of the winged-destined morph (*Figure 3H*), and the highest expression occurred in the thorax (*Figure 3I*), suggesting that *REPTOR2* was essential in activating autophagic degradation of the wing disc. To further support this notion, we fed newborn nymphs with *dsREPTOR2-RNA* and observed reduced *REPTOR2* transcripts in the wing disc and whole body by 53% and 65%, respectively (*Figure 3K and L*), as well as decreased gene expressions of *ATG13* and *ATG14* (*Figure 3J*). Knockdown of *REPTOR2* attenuated autophagy in the wing disc (*Figure 3M*), resulting in higher proportion of the winged morph, from 39% to 51% (*Figure 3N*).

## The effect of silencing TOR on wing disc autophagy and the proportion of the winged morph

Suppression of TOR is known to induce autophagic cell death during tissue degeneration in insects (*Berry and Baehrecke, 2007*; *Denton et al., 2012*). As expected in the wing disc, the fluorescence in situ hybridization (FISH) data indicated that *TOR* had higher abundance in winged- than wingless-destined morphs (*Figure 4A*). Feeding newborn nymphs with *dsTOR-RNA* reduced the *TOR* transcripts in the whole body and the wing disc by 48% and 34%, respectively (*Figure 4B and C*), and also declined winged proportion from 69% to 52% (*Figure 4D*). By contrast, activation of TOR via pharmacological agonist MHY1485 suppressed autophagy in the wing disc (*Figure 4E*) and increased winged proportion from 37% to 55% (*Figure 4F*). Since availability of amino acids is an important factor regulating TORC1 activity, depletion of dietary amino acids was used in the current study to suppress the TOR signaling pathway in pea aphids. Suppression of TOR via depletion of amino acids by 50% enhanced autophagy in wing disc (*Figure 4G*), and decreased winged proportion from 49% to 15% (*Figure 4H*).

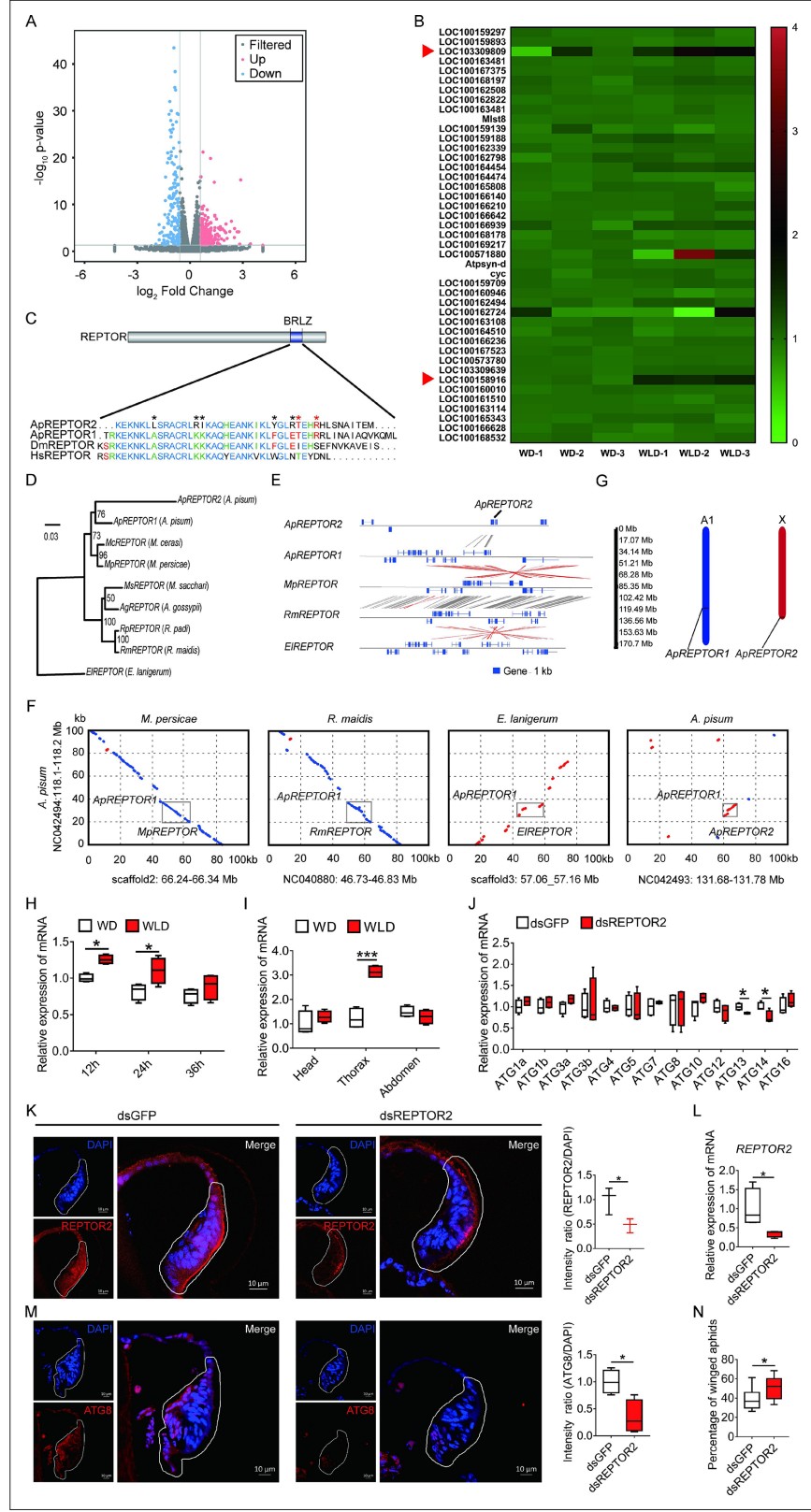

**Figure 3.** Pea aphid-specific *REPTOR2* is highly expressed in wingless-destined morph that activates autophagy in the wing disc. (**A**) The volcano plot of log-transformed FPKM showing the number of differentially expressed genes (DEGs) in wingless- versus winged-destined aphids at 24-h postbirth. Three biological replications were conducted in the RNA-seq analysis. (**B**) Transcript mapping to TOR signaling pathway of pea aphids and DEGs

*Figure 3 continued on next page*

*Figure 3 continued*

in two wing morphs. Red arrows indicated two DEGs, *REPTOR2* (LOC103309809) and *sodium-coupled neutral amino acid transporter 9* (LOC100158916), in wingless- versus winged-destined aphids. (**C**) Homology analysis of pea aphid REPTOR2. The amino acid sequence of conserved basic region leucine zippers (BRLZ) domain of pea aphid REPTOR2 was aligned with pea aphid REPTOR1 (ApREPTOR1), human REPTOR (HsREPTOR) and *Drosophila* REPTOR (DmREPTOR). Red and black asterisks indicated pea aphid-specific variation and ApREPTOR2-specific variation in amino acid, respectively. (**D**) An unrooted phylogenetic tree was constructed by the IQ-TREE method with coding sequence alignments for aphid REPTOR. The sequences of REPTOR identified from eight aphid species as well as the candidate pea aphid *REPTOR2* were compared. (**E**) Sequence comparisons of the LOC103309809 (*ApREPTOR2*) locus and its homologous regions in aphids. Homologous sequences were linked by gray dashes corresponding to the positive strand and by red dashes corresponding to the negative strand. (**F**) Gray boxes highlight the similarity of aphid REPTOR across 100 kb covering the REPTOR loci. (**G**) Chromosomal distribution of two REPTOR genes in pea aphids. (**H**) Relative expression levels of *REPTOR2* in first instar nymph at 12-, 24-, and 36-h postbirth in winged- and wingless-destined aphids (n=4). (**I**) Expression levels of *REPTOR2* in head, thorax and abdomen of winged- and wingless-destined aphids at 24-h postbirth (n=4). (**J**) Effects of ds*REPTOR2* on the expression of genes related to autophagy in whole body of the first instar nymphs (n=4). (**K**) Knockdown of *REPTOR2* reduced its expression in wing disc at 24-h postbirth. Fluorescence in situ hybridization (FISH) was used to detect the *REPTOR2* expression level in aphid wing discs. *REPTOR2* was hybridized with 5-CY3 in red, and nuclei were stained with DAPI in blue (n=3). (**L**) ds*REPTOR2* feeding decreased the gene expression of *REPTOR2*, as determined by qPCR (n=4). (**M**) Knockdown of *REPTOR2* resulted in the decline of autophagy in the wing disc at 30-h postbirth. Autophagy was noted by ATG8 (red) in immunofluorescence, and nuclei were stained by DAPI (blue) (n=4). (**N**) ds*REPTOR2* treatment increased the proportion of winged aphids (n=10). Wing discs shown in confocal were highlighted by white circles. WD, winged-destined; WLD, wingless-destined. Boxes show the interquartile range, and the line is the median value of each group. Independent sample t test was used to compare mean, and significant differences between treatments are indicated by asterisks: *p<0.05, **p<0.01, ***p<0.001.

The online version of this article includes the following source data and figure supplement(s) for figure 3:

**Source data 1.** Related to *Figure 3A*.

**Source data 2.** Related to *Figure 3B*.

**Source data 3.** Related to *Figure 3H*.

**Source data 4.** Related to *Figure 2I*.

**Source data 5.** Related to *Figure 3J*.

**Source data 6.** Related to *Figure 3K*.

**Source data 7.** Related to *Figure 2L*.

**Source data 8.** Related to *Figure 3M*.

**Source data 9.** Related to *Figure 2N*.

**Figure supplement 1.** *REPTOR1* was not involved in activation of autophagic degradation in the wing disc of the winged-destined morph.

**Figure supplement 1—source data 1.** Related to *Figure 3—figure supplement 1A*.

**Figure supplement 1—source data 2.** Related to *Figure 3—figure supplement 1B*.

**Figure supplement 1—source data 3.** Related to *Figure 3—figure supplement 1C*.

**Figure supplement 1—source data 4.** Related to *Figure 3—figure supplement 1D*.

**Figure supplement 1—source data 5.** Related to *Figure 3—figure supplement 1E*.

## Effects of co-suppression of *TOR* and *REPTOR2* on wing disc autophagy and proportion of the winged morph

To establish the causal link between *REPTOR2* and *TOR*, both genes were silenced via dsRNA feeding. Interestingly, aphids displayed lower wing disc autophagy and had higher winged proportion than ds*TOR* aphids (*Figure 4I and J*). Furthermore, ds*TOR* aphids showed increased *REPTOR2* expression in the wing disc and the whole body by 1.97- and 1.75-fold, respectively (*Figure 4K and L*), suggesting that *REPTOR2* was downstream of *TOR* in the signaling pathway.

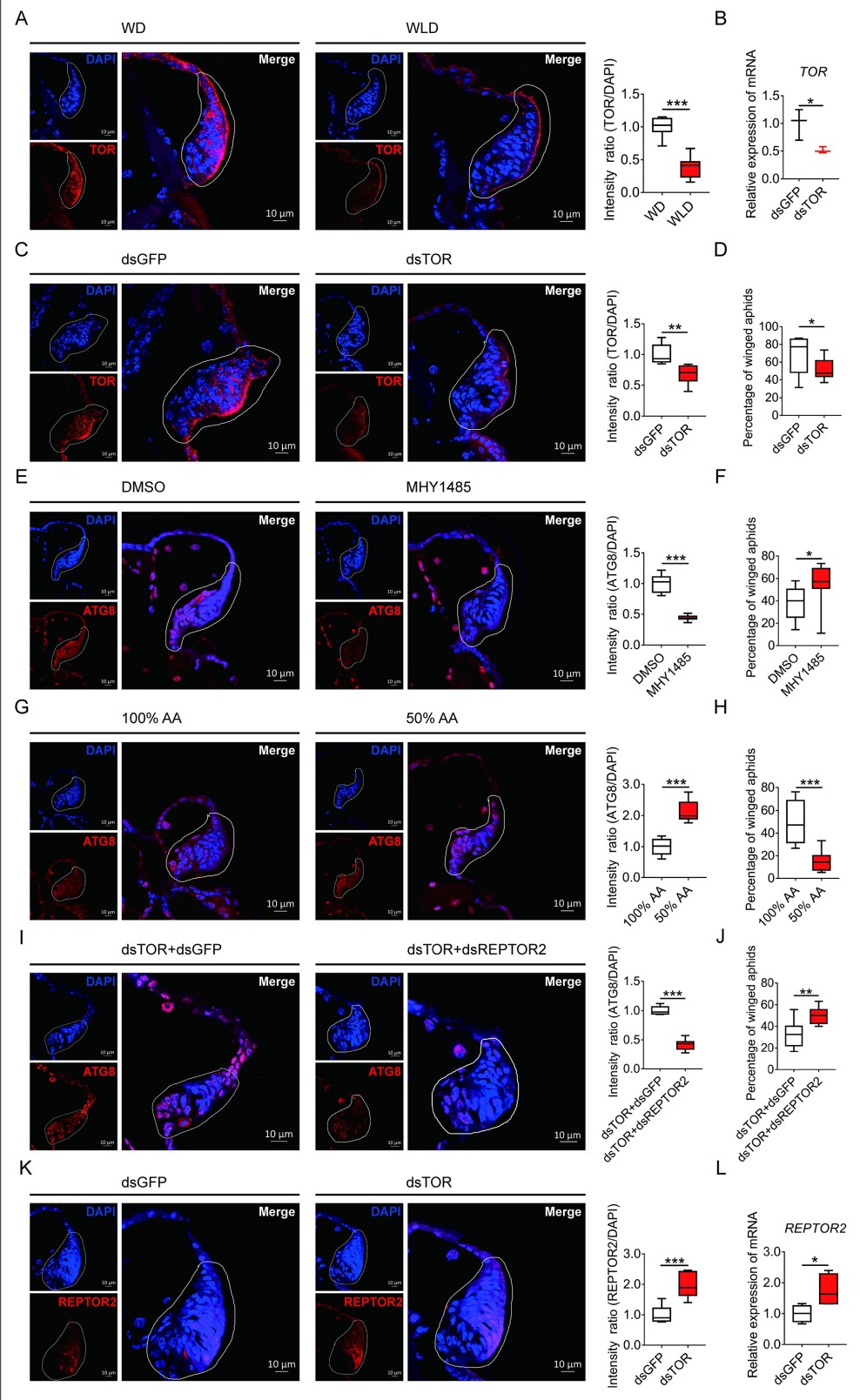

**Figure 4.** TOR negatively regulated autophagy in the wing disc and positively affected the proportion of winged aphids. (**A**) *TOR* expression in the wing disc of first instar nymph of winged- and wingless-destined aphids at 24-h postbirth, as determined by mRNA-FISH. *TOR* was hybridized with 5-CY3 in red, and nuclei were stained with DAPI in blue (n=9). (**B**) Newborn nymphs fed with dsRNA reduced *TOR* expression in whole body 24-h postbirth,

*Figure 4 continued on next page*

*Figure 4 continued*

as determined by qPCR (n=3). (**C**) Knockdown of *TOR* reduced its expression in the wing disc 24-h postbirth, shown in mRNA-FISH (n>7), and (**D**) increased the proportion of winged aphids (n=10). (**E**) Application of TOR agonist MHY1485 attenuated autophagy in the wing disc 30-h postbirth (n=9). (**F**) Activation of TOR increased the proportion of winged aphids (n>8). (**G**) Amino acids supplied at 50% of the standard diet (50% AA) activated autophagy in the wing disc 30-h postbirth (n=8). (**H**) 50% AA reduced the proportion of winged aphids (n=8). Autophagy in the wing disc was indicated by the hallmark protein ATG8 (red) in immunofluorescence. The nuclei (blue) were stained by DAPI. (**I**) Knockdown of both *TOR* and *REPTOR2* attenuated autophagy in the wing disc 30-h postbirth, relative to the control, ds*TOR*+ds*GFP* (n=7), and (**J**) increased the proportion of winged aphids (n=8). (**K**) Knockdown of *TOR* increased *REPTOR2* expression in the wing disc 24-h postbirth, shown in mRNA-FISH (n=7). *REPTOR2* was hybridized with 5-CY3 in red, and nuclei were stained with DAPI in blue. (**L**) Newborn nymphs fed with ds*TOR*-RNA increased *REPTOR2* expression in whole body 24-h post-birth, as determined by qPCR (n=4). WD, winged-destined; WLD, wingless-destined. Relative intensity was quantified by ImageJ. Wing discs shown in confocal was highlighted by white circles. Boxes show the interquartile range, and the line is the median value of each group. Independent sample t test was used to compare mean, and significant differences between treatments are indicated by asterisks: *p<0.05, **p<0.01, ***p<0.001.

The online version of this article includes the following source data for figure 4:

**Source data 1.** Related to *Figure 4A*.

**Source data 2.** Related to *Figure 4B*.

**Source data 3.** Related to *Figure 4C*.

**Source data 4.** Related to *Figure 4D*.

**Source data 5.** Related to *Figure 4E*.

**Source data 6.** Related to *Figure 4F*.

**Source data 7.** Related to *Figure 4G*.

**Source data 8.** Related to *Figure 4H*.

**Source data 9.** Related to *Figure 4I*.

**Source data 10.** Related to *Figure 4J*.

**Source data 11.** Related to *Figure 4K*.

**Source data 12.** Related to *Figure 4L*.

## Discussion

The wing disc consists of undifferentiated and proliferating cells designated to develop into a full insect wing during the nymphal stage (*Neto-Silva et al., 2009*). Fate of the wing disc is presumably determined by maternal signals triggered by environmental signals such as population density, physical contact, and the presence of natural enemies, as wing dimorphism in aphids is transgenerational (*Braendle et al., 2006*). It has been reported that ecdysteroids and miR-9b are the molecules that transduce maternal crowding signals to embryos, leading to a high proportion of winged offspring (*Vellichirammal et al., 2017*; *Shang et al., 2020*). However, few studies have focused on the role of the wing disc degradation at the onset of the nymphal stage in forming wing dimorphism. In this study, we demonstrated that a novel gene *REPTOR2*, highly expressed in the first instar nymphs of wingless-destined pea aphid morph 12-h postbirth, activated autophagy in the wing disc and resulted in its degeneration as early as 30-h postbirth. Different types of PCD have been reported to activate in tissue degeneration in many insect orders (*Xu et al., 2019*). Wing discs of soldier ant and some sex-dependent wingless morphs of lepidopterans are degenerated by apoptotic cell death (*Fujiwara and Ogai, 2001*; *Niitsu, 2001*; *Sameshima et al., 2004*; *Rajakumar et al., 2012*; *Rajakumar et al., 2018*). On the other hand, apoptosis did not seem to be necessary in wing disc degradation in aphids. Instead, typical features of autophagic cell death occurred at 30-h postbirth in wingless-destined morphs. Our results supported a model concerning aphid wing disc degeneration and development: Once experienced crowding, maternal aphids increased TOR expression in first instar nymphs. This was followed by the repression of REPTOR2 and autophagic degradation in the wing disc, thereby causing a high proportion of winged offspring. For the solitary maternal group on the other hand, TOR activity was arrested in the first instar offspring, and REPTOR2 and autophagy were subsequently activated in wing disc, leading to a high proportion of wingless offspring (*Figure 5*). Given the high

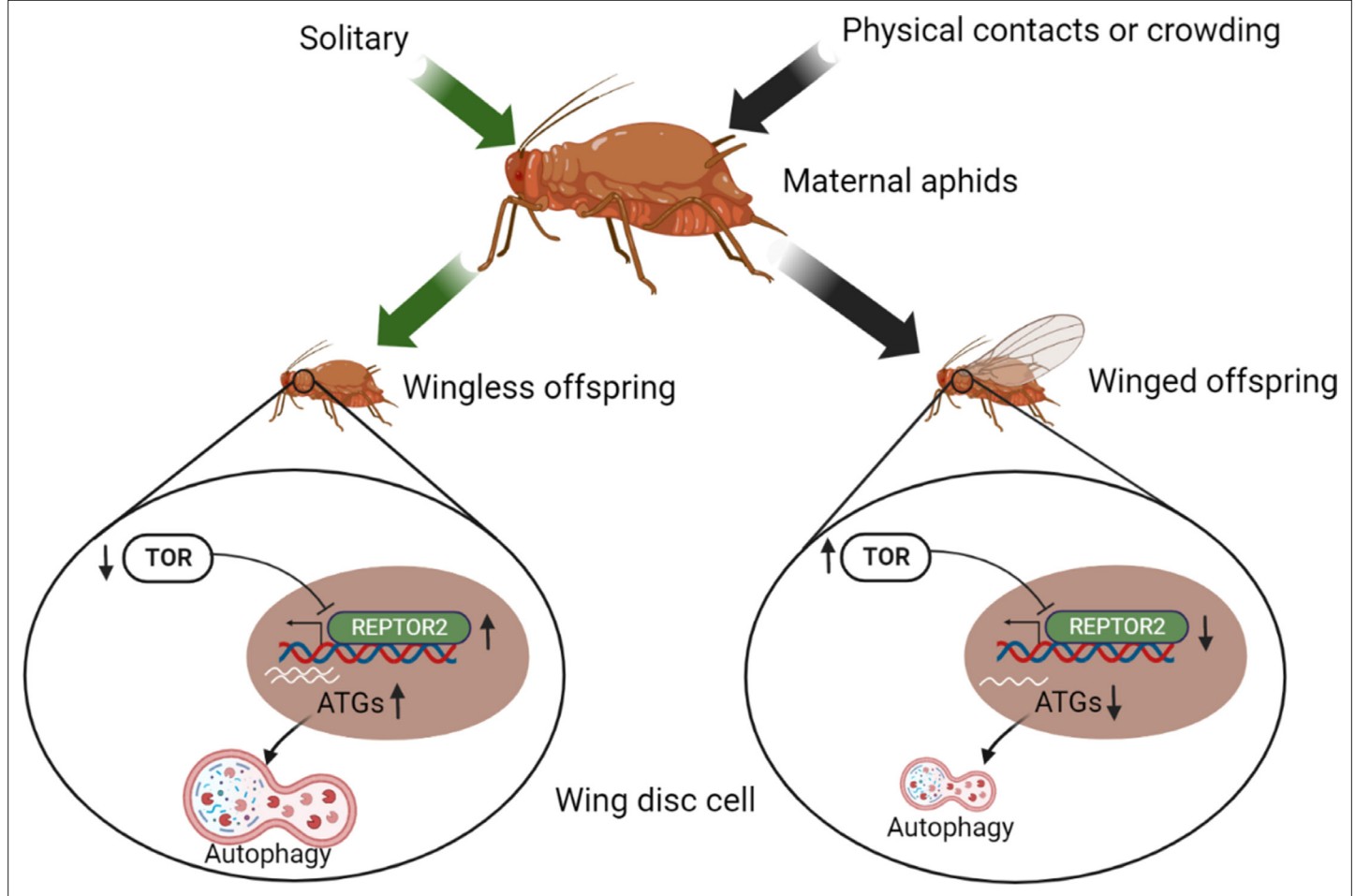

**Figure 5.** Illustration of the TOR signaling pathway-controlled wing disc degeneration of the first instar pea aphid nymphs. All treatments, indicated along the top, were applied to wingless maternal aphids. Arrows correspond to the treatments, coded by different colors. ATG, autophagy related gene; REPTOR2, repress by TOR 2; TOR, target of rapamycin. This graph was created with BioRender.com.

expression in thorax of wingless-destined morph at early first instar nymphal stage, it was likely that REPTOR2 transduced the TOR signaling to activate autophagic degeneration of the wing disc in a tissue-specific manner.

The TOR signaling pathway has been reported to modulate developmental plasticity in some insects (*Koyama et al., 2013*). For instance, activation of TOR activity in honeybees is responsible for the developmental transition from facultatively sterile workers to reproductive queens, while repressing TOR activity could induce worker characters in queen-destined individuals (*Michalak et al., 2007*). Transcriptomic analysis in maternal crowding of pea aphids suggested that genes functionally related to TOR signaling pathway were differentially expressed in mother aphids (*Parker et al., 2021*). We therefore hypothesized that the canonical pathway of TOR-REPTOR-autophagy may be responsible for developmental plasticity of the wing disc in aphids, in accordance with lower TOR abundance in wingless-destined morph. Suppression of TOR in *Drosophila* upregulates the expression of autophagy-related genes via phosphorylation of Ser527 and Ser530 in REPTOR (*Tiebe et al., 2015*). Similarly, pea aphid *REPTOR2* is predicted to contain a conserved BRLZ domain that is capable of binding to GTAAACAA, the binding motif of FOXO as well (*Tiebe et al., 2015*). FOXO is a transcription factor that relays the insulin signaling to insect wing dimorphism. In the planthopper, activating insulin signaling transforms short-winged to long-winged morph via repressing the FOXO activity (*Xu et al., 2015*). Activation of FOXO inhibits the development of the long-winged morph by suppressing vestigial (Vg) expression during the wing-morph decision-making stage (*Zhang et al., 2021*). It appears that FOXO and REPTOR2, respectively, relay the insulin and TOR signaling cascades

to control wing dimorphism in planthoppers and aphids. Alternatively, the BRLZ domain is required to form REPTOR/REPTOR-BP homo/hetero-dimer complex which conveys negative regulation of TOR signaling on genes related to autophagy (*Tiebe et al., 2015*). Our results are consistent with the canonical pathway regulated by TOR in that either knockdown of *REPTOR2* or activation of TOR could attenuate autophagy in the wing disc. These results suggest that *REPTOR2* from pea aphids served as a transcription factor to exert a negative effect on TOR-regulated autophagy initiation.

Insulin signaling is known to determine the wing-morph switch in planthoppers (*Xu et al., 2015*) and is also responsible for production of winged offspring regulated by miR-9b-ABCG4 cascade in brown citrus aphid (*Shang et al., 2020*). Furthermore, insulin could enhance TOR signaling by releasing the negative feedback of Akt on the tuberous sclerosis proteins TSC1 (hamartin) and TSC2 (tuberin) (*Blagosklonny et al., 2009*). It has been shown that embryos derived from pea aphids subjected to crowding treatment have higher insulin signaling (*Grantham et al., 2019*). This may lead to stronger TOR signaling in the first instar offspring, which could suppress wing disc autophagy and increase the proportion of winged aphids.

Gene duplication is a major source for novel gene production (*Kaessmann, 2010*; *Prince and Pickett, 2002*), which has become an important mechanism regulating phenotypic plasticity (*Chen et al., 2013*), including manipulating wing polyphenism in insects. For instance, a duplicated *fs3* (follistatin 3) determined the wingless morph formation for male pea aphids (*Li et al., 2020*). Another duplicated gene *InR2* (insulin receptor 2) was highly expressed in the wing bud of the planthopper that served as a regulatory switch of short-winged morph formation (*Xu et al., 2015*). Similarly, different from other seven aphid species with single copies of REPTOR, pea aphids possess two REPTOR genes. Although it has lost 2 exons, *ApREPTOR2* has higher similarity in coding sequence with *ApREPTOR1*, but shares little similarity in the flanking regions. This is in agreement with the chromosomal distribution results indicating that *ApREPTOR1* is located in A1 chromosome while *ApREPTOR2* is in X chromosome. *ApREPTOR2* is likely a duplicate from *ApREPTOR1*, followed by translocating from A1 to X chromosome. This may also warrant the reproductive role of sexual female in pea aphids because most sexual females are monomorphic wingless, whereas males that contain just single X chromosome are dimorphic (*Brisson, 2010*).

In the aphid wing polyphenism, differential expression of diverged paralogous genes could represent an adaptive response to particular environmental challenges (*Shigenobu et al., 2010*; *Brisson et al., 2010*). Evidently, *ApREPTOR2*-regulated wing disc degradation is significant for the winged-versus wingless-destined aphids, whereas the expression of its paralog *ApREPTOR1* was unaffected by wing morphs or tissue types (*Figure 3—figure supplement 1A*). Knockdown of *ApREPTOR1* did not affect the expression of *ApREPTOR2* and *ATGs* and the proportion of winged aphids (*Figure 3—figure supplement 1B–E*). These results indicated that *ApREPTOR2* rather than the canonical *REPTOR* activated a tissue-specific degradation in the wing disc and not a systemic PCD. Apparently, more research is necessary to further elucidate how other aphid species that contain a single *REPTOR* regulate autophagic degradation in the wing disc.

## Materials and methods

### Aphid rearing

The pea aphid *A. pisum* (strain: Ningxia Red) was originally collected from *Medicago sativa* in Ningxia Province and had been reared in the laboratory for over 8 years. Nymphs from the same parthenogenetic pea aphid female were reared on *Vicia faba* at 18–20°C, with 60% relative humidity and a photoperiod of a 16 hr: 8 hr (light/dark) cycle. To eliminate the transgenerational effects on offspring morphs, females were maintained at low density (three per plant) on *V. faba* seedlings for more than three generations (*Sutherland, 1969*). Other three strains of *A. pisum*, including Ningxia Green, Gansu Red, and Gansu Green, were originally collected from *M. sativa* in Ningxia Province and Gansu Province, respectively, and were used to test the inductive effect of two-adult contacts on the proportion of winged offspring.

### Induction of winged- and wingless-destined morphs

To effectively induce a high proportion of winged offspring, the maternal density and contacting duration were evaluated within petri dishes (35 mm in diameter) as described by *Sutherland, 1969*. Four

groups of aphids consisting of 1, 2, 4, or 8 wingless adults, respectively, were placed in a petri dish for 8-hr. Later, the two wingless adult groups were placed in a petri dish for 2-, 4-, and 8-hr, respectively. After treatment, each adult was then transferred to a freshly detached *V. faba* leaf kept in petri dishes with 1% agar, and allowed to reproduce for 24-hr. After newborns reached the fourth instar, the proportion of winged offspring from each adult was calculated. Furthermore, oviposition of each adult from the two wingless adult groups with 4-hr contact was documented for 5 days. Offspring were transferred daily to detached *V. faba* leaves that were changed every 3 days. Winged or wingless of each offspring was counted until the fourth instar. To eliminate the strain-specific induction, aphid strains of Ningxia Green, Gansu Red, and Gansu Green were also tested using two wingless adults contacting for 4-hr.

## Developmental time and wing morphology of winged- and wingless-destined morphs

Newly born nymphs of winged- or wingless-destined morphs were raised individually on fresh, detached leaf of *V. faba* in petri dishes. The nymphal developmental time and the pre-reproductive period were recorded every 12-hr. The digital microscope (Keyence VHX-5000, Osaka, Japan) was used to determine the wing morphology in nymphs of each instar.

## Histology

Winged- and wingless-destined morphs at 24-, 30-, and 36-hr postbirth were collected and placed in 4% paraformaldehyde fixative overnight at 4°C, washed in phosphate-buffered saline for three times and dehydrated at 4°C for 24-hr in 30% sucrose solution. After freezing at –20 °C in Tissue Freezing Medium (Leica), aphids were sectioned at 10 μm using a Leica CM1950 freezing microtome, and stained with H&E (Beyotime, C0105) according to the standard protocol. Images of wing discs were taken using a Nikon light microscope, equipped with a DS-Fi1c camera (Nikon), and the images were generated using the NIS-Element D software (Nikon). The relative area of wing disc was analyzed using ImageJ software.

## Transmission electron microscope

Since wing disc degeneration initiated at 30-hr postbirth, winged- and wingless-destined morphs at this time point were collected and fixed with 2.5% (vol/vol) glutaraldehyde, and washed four times in PB (0.1 M, pH 7.4). Aphids were then fixed with 1% (wt/vol) osmium tetraoxide in PB for 2-hr at 4°C, and dehydrated through a graded ethanol series (30%, 50%, 70%, 80%, 90%, 100%, and 100%, 7 min each) into pure acetone (2×10 min). Samples were infiltrated in a graded mixture (3:1, 1:1, and 1:3) of acetone and SPI-PON812 resin 16.2 g SPI-PON812, 10 g dodecyl succinic anhydride, and 8.9 g N-methylolacrylamide, then in pure resin. Aphids were embedded in pure resin with 1.5% benzyl dimethyl amine and polymerized for 12-hr at 45°C, and 48-hr at 60°C. The ultrathin sections (70 nm thick) were sectioned with microtome (Leica EM UC6, Austria), and then double-stained by uranyl acetate and lead citrate, and imaged under transmission electron microscope (FEI Tecnai Spirit 120kV, USA). Ultrathin sections were used to count the cell number that had signs of autophagy (mitophagy, cytoplasmic vacuoles, large autophagosomes sequestering visible remnants of organelles, and membranous whorls) (*Simonet et al., 2018*). The percentage of autophagic cells was determined by randomly counted cells in the wing disc.

## Immunofluorescence detection and TUNEL assays

The frozen sections of aphids with different treatments were rinsed three times in TBST (TBS with 0.05% Tween-20). TUNEL assays were carried out by One Step TUNEL Apoptosis Assay Kit (Beyotime, C1089) according to the manufacturer's protocol. For immunofluorescence, frozen sections of aphids were blocked with SuperBlock T20 (Pierce) for 20 min. The samples were incubated with the primary antibody (anti-ATG8, 1:200) overnight at 4°C, rinsed three times in TBST, and incubated with the secondary antibody (1:500) at room temperature for 2-hr. A negative control was performed in each independent experiment. The samples were rinsed for three times in TBST, and mounted in Fluoroshield Mounting Medium with DAPI (Abcam). Sections were imaged using a Zeiss LSM710 confocal microscope (Zeiss, Germany). The polyclonal antibody rabbit anti-ATG8 was kindly provided by Professor Le Kang (Institute of Zoology, Chinese Academy of Sciences). Anti-rabbit IgG (H+L),

F(ab')2 Fragment (Alexa Fluor 594 Conjugate, 8889S) was purchased from CST. Relative intensities of TUNEL and ATG8 were analyzed using the ImageJ software.

## Fluorescence in situ hybridization

FISH was performed with the technique modified by *Kliot et al., 2014*. After washing by TBST (TBS with 0.2% Triton-X) for 10 min, the frozen sections were rinsed for three times in hybridization buffer (20 mM Tris-HCl, pH 8.0, 0.9 M NaCl, 0.01% [wt/vol] sodium dodecyl sulfate, and 30% [vol/vol] formamide) for pre-hybrid (without the probe). The frozen sections of aphids were then hybridized overnight in hybridization buffer containing 10 pmol of the fluorescent RNA probe (conjugated with Cy3). Afterward, the frozen sections were rinsed for three times in TBST, and mounted in Fluoroshield Mounting Medium with DAPI (Abcam). Sections were imaged using a Zeiss LSM710 confocal microscope (Zeiss, Germany).

## Quantitative PCR

Total RNA of aphid samples was extracted by TRIzol Reagent (Thermo Fisher) according to the manufacturer's protocol, and cDNA was synthesized using the FastQuant RT Kit with gDNase (Tiangen). The qPCR reactions were performed with the PowerUp SYBR Green Master Mix (Applied Biosystems). RT-qPCR reactions were carried out on the QuantStudio 12 K Flex Real-Time PCR System (ABI, A25742) as follows: 30 s at 95°C; followed by 40 cycles of 10 s at 95°C, 30 s at 60°C; and finally one cycle of 15 s at 95°C, 60 s at 60°C, and 15 s at 95°C. The melting curves were used to determine the specificity of the PCR products. The housekeeping gene *actin* (ACYPI000064) was used as the internal qPCR standard to analyze gene expression (*Simonet et al., 2018*). The relative level of each target gene was standardized by comparing the copy numbers of target mRNA with copy numbers of *actin*. Data were analyzed by the $2^{-\Delta\Delta CT}$ relative quantification method. Specific primers for each gene were designed from the aphid sequences using Primer Premier 6 software. The availability of each pair of primers had been tested in the preliminary experiments. The oligonucleotides are listed in *Supplementary file 1*.

We tested for enrichment of expression of *REPTOR1* or *REPTOR2* in aphid heads, thorax, and abdomen using qPCR. Aphid dissection was performed with a technique modified by *Chen et al., 2015*. Winged- and wingless-destined aphids at 24-hr postbirth were collected in 2 ml RNAlater (Thermo Fisher Scientific) and kept at 4°C overnight, and 100 µl 0.1% PBST was added to help RNAlater penetration. Head, thorax, and abdomen from around 80 aphids were dissected in cold RNAlater. RNA extraction and qPCR of dissected tissues were carried out as above.

## Pharmacological experiments and RNA interference via feeding

Twenty newborn aphids were placed in cylindrical containers, 20 mm diameter × 20 mm height. Each container was provided with a 100 µl artificial diet (*Haribal and Jander, 2015*). The autophagy inhibitor 3-MA (Selleck, S2767) was dissolved in double distilled water and incorporated into diet at 10 µM. The autophagy agonist rapamycin (Selleck, S1039) was dissolved in ethanol and mixed in diet at a concentration of 10 µM, and an equivalent amount of ethanol was used as a feeding control. The TOR agonist MHY1485 (Selleck, S7811) was dissolved in DMSO and mixed in diet at a concentration of 10 µM, and an equivalent amount of DMSO was used as a feeding control. Diet with 50% amino acids (each amino acid was supplied at 50% of standard formulation) was used to repress TOR activity, and a normal diet was used as a control. All chemicals were administered via 24-hr feeding.

The T7 RiboMAX Express RNAi System (Promega, P1700) was used to synthesize double-stranded RNA (dsRNA) according to the manufacturer's protocol. dsRNAs, that is, dsREPTOR1, dsREPTOR2, dsTOR (5 µl, 10 µg/µl), or dsGFP control were added to 95 µl artificial diet to feed aphids. The interference efficiencies were measured by qPCR.

## Co-silencing of *TOR* and *REPTOR2*

Newborn nymphs were fed with a dsRNA mixture of ds*TOR*+ds*REPTOR2*, and ds*TOR*+ds*GFP* served as the control (50 µg for each dsRNA). Nymphs at 30-hr postbirth were collected for immunofluorescence detection. In addition, each winged or wingless aphid was counted until the fourth instar.

## RNA-seq and data analysis

To determine DEGs in winged- and wingless-destined morphs, the whole body of 20 aphids at 24-hr postbirth was collected for further RNA extraction. RNA purity and quantification were evaluated using

the NanoDrop 2000 spectrophotometer (Thermo Fisher Scientific). RNA integrity was assessed using the Agilent 2100 Bioanalyzer (Agilent Technologies). The filtered data were obtained by removing low-quality reads from the raw data with HTseq-count software (*Anders et al., 2014*). After quality control, paired-end clean reads were aligned to the reference genome downloaded from NCBI (ftp://ftp.ncbi.nlm.nih.gov/genomes/all/GCF/005/508/785/GCF_005508785.1_pea_aphid_22Mar2018_4r6ur/GCF_005508785.1). FPKM of each gene was calculated using Cufflinks (v2.1.1). Differential expression analysis was performed using the DESeq (2012) R package (*Anders and Huber, 2012*). $\log_2$ |fold change|>0.58, p value<0.05 was set as the threshold for significantly differential expression. Sequence data have been deposited in the National Center for Biotechnology Information's Sequence Read Archive, https://www.ncbi.nlm.nih.gov/sra (accession no. PRJNA857589).

## Analysis of REPTOR

Two *REPTOR* genes were identified in the *A. pisum* reference genome (GCF_005508785.1): LOC100168197 (*REPTOR1*) and LOC103309809 (*REPTOR2*).

The REPTOR homologs in the *M. cerasi*, *M. persicae*, *M. secchari*, *A. gossyppii*, *R. padi*, *R. maidis*, and *E. lanigerum* were identified by BLASTp using pea aphid REPTOR1. Coding sequences of homologous genes of REPTOR, including LOC100168197 (*REPTOR1*, *A. pisum*), LOC103309809 (*REPTOR2*, *A. pisum*), Mca07599.t1 (*M. cerasi*), g26487.t1 (*M. persicae*), LOC112600553 (*M. secchari*), LOC114124537 (*A. gossyppii*), g1773.t1 (*R. padi*), LOC113558611 (*R. maidis*), and jg23767.t1 (*E. lanigerum*), were obtained from NCBI and AphidBase. Sequences were aligned with MUSCLE v3.8.1551 (*Edgar, 2004*). The phylogenetic analysis was performed in IQ-TREE v2.1.4-beta with HKY+F+G4 model (*Minh et al., 2020*) and the phylogenetic tree was visualized in FigTree v1.4.4 (http://tree.bio.ed.ac.uk/software/figtree/). Genomic sequences of orthologous regions of REPTOR were obtained from *M. persicae* (v2.0), *R. maidis* (v1.0), and *E. lanigerum* (v1.0), which have chromosome-level genome assemblies available at Aphidbase. Sequences were compared with NCBI blast v2.2.26 and homologous sequences were linked by the gray color when sequences matched to positive strand and by the red color when sequences matched to negative strand. Dot plots were generated using MUMmer v3.23. MapGene2Chrom v2.1 was used to analyze the chromosomal distribution of two pea aphid REPTOR genes (http://mg2c.iask.in/mg2c_v2.1/). Conserved domains of REPTOR were identified by SMART (http://smart.embl-heidelberg.de/). The amino acid sequences of REPTOR homologs from *A. pisum*, *D. melanogaster*, and *H. sapiens* were downloaded from GenBank and aligned with *ApREPTOR2* using MUSCLE v3.8.1551.

## Statistical analysis

Statistical analyses were performed with SPSS software (Chicago, IL) and GraphPad software. All data were checked for normality by the Shapiro-Wilk test. Student's t tests were used to analyze the relative area of the wing disc, percentage of autophagic cells, the number of autophagosomes per wing disc, intensity ratios of ATG8/DAPI, TUNEL/ DAPI, REPTOR2/DAPI, and TOR/ DAPI, percentage of winged aphids, and quantification of gene expression for two-group comparisons. ANOVA was used to analyze the effects of maternal density and contacting duration on the proportion of winged offspring, and proportion of winged offspring in each day for 4-hr contacting treatment with two female adults.

## Acknowledgements

The authors thank our colleagues Prof. Jinfeng Chen for his support in genetic analysis and Prof. Chenzhu Wang for his advises in project coordination and manuscript writing. The authors appreciate TEM sample preparing and analysis support from Can Peng, Center for Biological Imaging, Institute of Biophysics, CAS. This project was supported by the National Key R&D Program of China (No. 2022YFD1400800), the Strategic Priority Research Program of the Chinese Academy of Sciences (No. XDPB16), and the National Natural Science Foundation of China (Nos. 32250002, 31970453, and 31870394).

## Additional information

### Funding

| Funder | Grant reference number | Author |
|--------|------------------------|--------|
| Ministry of Science and Technology of the People's Republic of China | the National Key R&D Program of China (no. 2022YFD1400800) | Yucheng Sun |
| Chinese Academy of Sciences | the Strategic Priority Research Program of the Chinese Academy of Sciences (no. XDPB16) | Yucheng Sun |
| National Natural Science Foundation of China | the National Natural Science Foundation of China (no. 32250002) | Yucheng Sun |
| National Natural Science Foundation of China | the National Natural Science Foundation of China (no. 31970453) | Yiyang Yuan |
| National Natural Science Foundation of China | the National Natural Science Foundation of China (no. 31870394) | Huijuan Guo |

The funders had no role in study design, data collection and interpretation, or the decision to submit the work for publication.

### Author contributions

Erliang Yuan, Conceptualization, Data curation, Writing – original draft; Huijuan Guo, Formal analysis, Funding acquisition, Writing – review and editing; Weiyao Chen, Yingjie Mi, Methodology; Bingru Du, Zhaorui Qi, Investigation; Yiyang Yuan, Data curation, Funding acquisition; Keyan Zhu-Salzman, Writing – review and editing; Feng Ge, Supervision, Writing – review and editing; Yucheng Sun, Conceptualization, Resources, Supervision, Funding acquisition, Project administration, Writing – review and editing

### Author ORCIDs

Erliang Yuan ⓘ http://orcid.org/0000-0001-7825-1908
Huijuan Guo ⓘ http://orcid.org/0000-0002-4432-6446
Yucheng Sun ⓘ http://orcid.org/0000-0003-2353-0218

### Decision letter and Author response

Decision letter https://doi.org/10.7554/eLife.83023.sa1
Author response https://doi.org/10.7554/eLife.83023.sa2

# Additional files

### Supplementary files

- Supplementary file 1. Primers used for the study.
- MDAR checklist

### Data availability

Sequencing data have been deposited in SRA under accession ID PRJNA857589.

The following dataset was generated:

| Author(s) | Year | Dataset title | Dataset URL | Database and Identifier |
|-----------|------|---------------|-------------|-------------------------|
| Yuan E | 2022 | Wing dimorphism in Acyrthosiphon pisum | https://www.ncbi.nlm.nih.gov/bioproject/PRJNA857589/ | NCBI BioProject, PRJNA857589 |

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
