## [Editor Report]

This study addresses the genetic basis of an iconic example of developmental plasticity. The background and hypothesis are clear, and the experiments are well conducted. The paper shows that winglessness in aphids involves autophagy (rather than apoptosis) of the wing primordium, and that this is regulated by a novel gene, REPTOR2, which is a target of TOR signaling.

---

## [Decision Letter]

**Decision letter after peer review:**

Thank you for submitting your article "A Novel Gene *REPTOR2* Activates the Autophagic Degradation of Wing Disc in pea aphid" for consideration by *eLife*. Your article has been reviewed by 2 peer reviewers, and the evaluation has been overseen by a Reviewing Editor and K VijayRaghavan as the Senior Editor. The reviewers have opted to remain anonymous.

Essential revisions:

1) The authors do not establish a direct link between REPTOR2 and TOR. New data is needed unambiguously show that suppression of TOR signalling works through REPTOR2 to induce autophagy of the wing primordium. Both reviewers suggest specific experiments that could be done, relating to measuring REPTOR2 expression in dsTOR animals and performing double knockdown experiments.

2) Clarifications are required about data structure and statistical analyses.

*Reviewer #1 (Recommendations for the authors):*

The methods utilized are appropriate for the study and generally sufficient to support the main findings of the paper. Nevertheless, I have a number of recommendations for how the manuscript and study can be improved.

1) A major conclusion of the study is that the "results suggest that REPTOR2 from pea aphids served as a transcription factor to exert a negative effect of TOR signaling on autophagy initiation" (L267). The model is therefore that suppression of TOR signaling works through REPTOR2 to induce autophagy of the wing primordium. However, as implied by Figure 5, the study does not unambiguously place REPTOR2 downstream of TOR in the regulation of wing dimorphism. To demonstrate this, the authors need to conduct an epistatic analysis, which appears to be straightforward given the tools utilized in the study. Specifically, the authors should show the effects of the suppression of TOR signaling in inducing autophagy of the wing primordium and production of wingless aphids, either by dsTOR.RNAI or amino acid deprivation (Figure 4 C, D, G, H), should be significantly reduced or eliminated when REPTOR1 is also suppressed by dsREPTOR2.RNAi (Figure 4 M, N). Such a demonstration, which could be achieved by feeding aphids both dsTOR.RNAI and dsREPTOR2.RNAi, would substantially strengthen the findings of the paper, and is, to my mind, an essential experiment.

2) While it seems likely that TOR activates REPTOR2 directly through phosphorylation (there is a physical association between TOR and REPTOR in *Drosophila*), the observation that REPTOR2 expression is elevated in WLD nymphs (Figure 3 H, I) means that there may also be a transcriptional response also. Demonstrating that REPTOR2 expression is elevated upon suppression of TOR signaling would strengthen a direct relationship between the two.

3) It is not clear how the data from Figure 1F were collected. The authors state that the winged and wingless morphs of A.pisum cannot be distinguished until the third instar, so they utilize a culturing technique to increase the proportion of WLD or WD nymphs to ~90% (Figure S1). Nevertheless, the data in Figure 1F showing the area of the wing primordia in WD versus WLD shows a 100% separation in the area between both morphs by 36h after birth. I am surprised that at least one of the nymphs raised in the WLD-inducing conditions did not show WD development, or one of the nymphs in the WD-inducing conditions did not show WLD development. The figure legend states that there were >16 replicates for each group so just by probability alone, at least one of the ostensibly WLD nymphs are actually winged, or one of the ostensibly WD nymphs are wingless. A back-of-an-envelope calculation, where the probability of generating a winged or wingless morph is 0.9, and there are 32 trials suggests the probability of getting 100% WLD or WD nymphs in each treatment is 0.034, which is unlikely (and below the significance threshold of 0.05).

4) The authors used a culture technique that generates ~50% winged/wingless morphs to explore how the knockdown of REPTOR2 and TOR affects the ratio of WLD and WD. While, within experiments, the results show significant changes in ratio with an experimental treatment, the percentage of winged aphids is highly variable among the controls, to the same extent as the observed effect size. For example in Figure 1I, 3-MA significantly increases the percentage of winged aphids from ~55% (control) to ~65% (treatment), but the control treatment for the rapamycin treatment (EtOH) in Figure 1H is also ~65%. Indeed, across the study, the percentage of winged aphids in the control groups range from ~40% (Figure 4F) to ~65% (Figure 1H). This variability is troubling, and while I don't feel that it compromises the results, the authors may consider using culture conditions that are more consistent, for example by conducting their knockdowns in the conditions where >90% of the offspring are expected to be one morph or the other.

5) Figure 3J: The hypothesis suggests that the knockdown of REPTOR2 should decrease the expression of autophagic genes since they are upregulated in WLD nymphs (Figure 2D). This is only true for two genes. The authors may like to speculate as to why they did not observe a robust transcriptional response to dsREPTOR2.RNAi treatment.

6) In general, the paper suffers from a lack of transparency with regard to statistical methods. In particular, while some of the bar charts include all the data (e.g. Figure 1F), almost all the others do not (e.g. all the figures showing mean percentage winged aphids). Although the sample sizes are provided in the legends, standard error bars are not sufficient to capture the true data. Box plots capture both means, but also the distribution of the underlying data, and would be more appropriate for this study. The authors should also provide all their data as a Supplementary file, not just the gene expression data.

*Reviewer #2 (Recommendations for the authors):*

In the introduction, I think a more thorough discussion of the Target of Rapamycin pathway would help. Currently, the connection between TOR and REPTOR is not clear.

In the results text, it would be useful to provide a bit more information about why you reduced amino acids in Figure 4G. I understand that you expect reducing the amino acid concentration of the diet to reduce TOR signalling, but this isn't explicitly stated when you introduce the experiment.

Finally, to show that your manipulations of TOR signalling occur through REPTOR2, and not through a parallel mechanism, it would useful to (1) examine REPTOR2 expression in the dsTOR animals and (2) perform the double knockdowns (dsREPTOR2 and dsTOR).

---

## [Author Response]

Essential revisions:1) The authors do not establish a direct link between REPTOR2 and TOR. New data is needed unambiguously show that suppression of TOR signalling works through REPTOR2 to induce autophagy of the wing primordium. Both reviewers suggest specific experiments that could be done, relating to measuring REPTOR2 expression in dsTOR animals and performing double knockdown experiments.

As suggested, experiments silencing both *TOR* and *REPTOR2* have been performed (Figure 4I-L). Our results showed that simultaneous knockdown of *TOR* and *REPTOR2* in aphids caused lower autophagy in the wing disc and higher proportion of the winged morph than single knockdown of *TOR* (Figure 4I and 4J). ds*REPTOR2* could reverse the positive effect of ds*TOR* on autophagy, suggesting that REPTOR2 is the key regulator downstream of TOR in the signaling pathway. In addition, we quantified in situ the *REPTOR2* expression in wing discs of ds*TOR* aphids vs. ds*GFP* aphids, and found that ds*TOR* aphids had higher REPTOR2 expression both in the wing disc and whole body than ds*GFP* aphids (Figure 4K and 4L). These results support our conclusion that suppression of TOR induces autophagy in the wing disc by activating REPTOR2, which leads to a lower proportion of winged aphids.

2) Clarifications are required about data structure and statistical analyses.

We have improved these issues as follow: (1) we updated some figures in the main text and supplementary files (Figure 1A-C, Figure 1F, Figure 2B-I, Figure 3H-N, Figure 4A-H, Figure 1—figure supplement 1, Figure 3—figure supplement 1) by adding Box plots as suggested by a reviewer. Means and the distribution of these data are now shown to address the transparency issue with regard to data structure. (2) Two additional experiments have been conducted to strengthen the connection between TOR and REPTOR2. The results showed that ds*TOR* in newborn nymphs increased the expression of *REPTOR2* in the wing disc and whole body by 1.97-fold and 1.75-fold, respectively (Figure 4K and 4L). Co-silencing of *TOR* and *REPTOR2* caused lower wing disc autophagy and higher winged proportion in aphids than silencing *TOR* alone. (3) We included the data collection procedure for Figure 1F (see No.3 response to reviewers #1 for details).

Reviewer #1 (Recommendations for the authors):The methods utilized are appropriate for the study and generally sufficient to support the main findings of the paper. Nevertheless, I have a number of recommendations for how the manuscript and study can be improved.1) A major conclusion of the study is that the "results suggest that REPTOR2 from pea aphids served as a transcription factor to exert a negative effect of TOR signaling on autophagy initiation" (L267). The model is therefore that suppression of TOR signaling works through REPTOR2 to induce autophagy of the wing primordium. However, as implied by Figure 5, the study does not unambiguously place REPTOR2 downstream of TOR in the regulation of wing dimorphism. To demonstrate this, the authors need to conduct an epistatic analysis, which appears to be straightforward given the tools utilized in the study. Specifically, the authors should show the effects of the suppression of TOR signaling in inducing autophagy of the wing primordium and production of wingless aphids, either by dsTOR.RNAI or amino acid deprivation (Figure 4 C, D, G, H), should be significantly reduced or eliminated when REPTOR1 is also suppressed by dsREPTOR2.RNAi (Figure 4 M, N). Such a demonstration, which could be achieved by feeding aphids both dsTOR.RNAI and dsREPTOR2.RNAi, would substantially strengthen the findings of the paper, and is, to my mind, an essential experiment.

According to your comments, we determined the wing disc autophagy and proportion of the winged morph when *TOR* and *REPTOR2* were both silenced in aphids. Knocking down both *TOR* and *REPTOR2* compromised wing disc autophagy and enhanced winged proportion in aphids relative to silencing *TOR* alone (Figure 4I and 4J), These results thus have strengthened our conclusion.

2) While it seems likely that TOR activates REPTOR2 directly through phosphorylation (there is a physical association between TOR and REPTOR in *Drosophila*), the observation that REPTOR2 expression is elevated in WLD nymphs (Figure 3 H, I) means that there may also be a transcriptional response also. Demonstrating that REPTOR2 expression is elevated upon suppression of TOR signaling would strengthen a direct relationship between the two.

As recommended, R*EPTOR2* expression was determined when newborn nymphs were fed with dsRNA of *TOR* or *GFP*. The results showed that ds*TOR* increased the transcripts of *REPTOR2* both in the wing disc and whole body by 1.97-fold and 1.75-fold (Figure 4K and 4L), suggesting that TOR could repress the transcriptional expression of *REPTOR2*.

3) It is not clear how the data from Figure 1F were collected. The authors state that the winged and wingless morphs of A.pisum cannot be distinguished until the third instar, so they utilize a culturing technique to increase the proportion of WLD or WD nymphs to ~90% (Figure S1). Nevertheless, the data in Figure 1F showing the area of the wing primordia in WD versus WLD shows a 100% separation in the area between both morphs by 36h after birth. I am surprised that at least one of the nymphs raised in the WLD-inducing conditions did not show WD development, or one of the nymphs in the WD-inducing conditions did not show WLD development. The figure legend states that there were >16 replicates for each group so just by probability alone, at least one of the ostensibly WLD nymphs are actually winged, or one of the ostensibly WD nymphs are wingless. A back-of-an-envelope calculation, where the probability of generating a winged or wingless morph is 0.9, and there are 32 trials suggests the probability of getting 100% WLD or WD nymphs in each treatment is 0.034, which is unlikely (and below the significance threshold of 0.05).

It is difficult to distinguish winged and wingless morphs of pea aphids morphologically until the third instar nymphal stage, so 20X light microscope was used to check the structure of wing disc at the first instar. Empirically, the typical wing discs of WD- or WLD-morph were then selected for further imaging and analysis. Thus, the area of the wing primordia of selected WD vs. WLD aphids showed a 100% separation 36h post birth. Furthermore, Figure 2—figure supplement 1 exhibited an insect culturing protocol for sampling 100% WLD-nymphs and ~90% WD-nymphs (Figure1A), which could also better help us to separate WLD- and WL-nymphs, and obtain the clean data on wing primordia of two morphs.

4) The authors used a culture technique that generates ~50% winged/wingless morphs to explore how the knockdown of REPTOR2 and TOR affects the ratio of WLD and WD. While, within experiments, the results show significant changes in ratio with an experimental treatment, the percentage of winged aphids is highly variable among the controls, to the same extent as the observed effect size. For example in Figure 1I, 3-MA significantly increases the percentage of winged aphids from ~55% (control) to ~65% (treatment), but the control treatment for the rapamycin treatment (EtOH) in Figure 1H is also ~65%. Indeed, across the study, the percentage of winged aphids in the control groups range from ~40% (Figure 4F) to ~65% (Figure 1H). This variability is troubling, and while I don't feel that it compromises the results, the authors may consider using culture conditions that are more consistent, for example by conducting their knockdowns in the conditions where >90% of the offspring are expected to be one morph or the other.

We appreciate these comments. Ideally, the control group should maintain a ~50% of the winged morph for the evaluations (Rapamycin, 3-MA or TOR dsRNA) to be unbiased. We followed the culture technique by Vellichirammal et al. (2017) and carefully repeated every batch of the control. However, we were unable to avoid variations between batches. Therefore, we focused our comparisons on differences between the treatment and control. In fact, this level of variation among controls is commonly seen in other publications, as listed below.

Parker BJ, Brisson JA. 2019. A Laterally Transferred Viral Gene Modifies Aphid Wing Plasticity. Current Biology, 29(12), 2098-2103.e2095.

Reyes ML, Laughton AM, Parker BJ, Wichmann H, Fan M, Sok D, Gerardo NM. 2019. The influence of symbiotic bacteria on reproductive strategies and wing polyphenism in pea aphids responding to stress. Journal of Animal Ecology, 88(4), 601-611.

Vellichirammal NN, Gupta P, Hall TA, Brisson JA. 2017. Ecdysone signaling underlies the pea aphid transgenerational wing polyphenism. Proceedings of the National Academy of Sciences, 114(6), 1419-1423.

5) Figure 3J: The hypothesis suggests that the knockdown of REPTOR2 should decrease the expression of autophagic genes since they are upregulated in WLD nymphs (Figure 2D). This is only true for two genes. The authors may like to speculate as to why they did not observe a robust transcriptional response to dsREPTOR2.RNAi treatment.

A number of autophagic genes are associated with autophagy process, differing in temporal and spatial expression pattern. For example, only two autophagic genes *Ulk1* and *ATG8* displayed oscillation expression throughout the light/dark cycle in mouse liver (Ma et al., 2011). Activating autophagy in response to starvation is highly conserved in eukaryotic cells. However, only one autophagic gene *Atg8a* was up-regulated in *Drosophila* fat body under starvation stress (Zinke et al., 2002; Juhasz et al., 2007). In the salivary gland and midgut autophagic cell death process, the expression of *Atg2*, *Atg4*, *Atg5*, *Atg7*, *Atg9* and *Atg12* were increased (Lee et al., 2003; Gorski et al., 2003). The autophagic genes could be regulated by other transcription factors. Similar with current study, only a few autophagic genes could be regulated. For example, ATG8 and ATG9 could be upregulated by transcription factor EB (TFEB) when HeLa cells were starved while ATG8, ATG1 and ATG5 were regulated by transcription factors E2F1 in DNA damage-induced autophagy (Settembre et al., 2011; Polager et al., 2008).

Reference:

Gorski SM, Chittaranjan S, Pleasance ED, Freeman JD, Anderson CL, Varhol RJ et al. A SAGE approach to discovery of genes involved in autophagic cell death. Curr Biol 2003; 13: 358–363.

Juhasz G, Puskas LG, Komonyi O, Erdi B, Maroy P, Neufeld TP et al. Gene expression profifiling identififies FKBP39 as an inhibitor of autophagy in larval *Drosophila* fat body. Cell Death Differ 2007; 14: 1181–1190.

Lee CY, Clough EA, Yellon P, Teslovich TM, Stephan DA, Baehrecke EH. Genome-wide analyses of steroid- and radiation-triggered programmed cell death in *Drosophila*. Curr Biol 2003; 13: 350–357.

Ma, D., Panda, S., Lin, J. D. Temporal orchestration of circadian autophagy rhythm by C/EBPβ. The EMBO Journal 2011; 30(22), 4642-4651.

Polager S, Ofir M, Ginsberg D. E2F1 regulates autophagy and the transcription of autophagy genes. Oncogene 2008; 27, 4860–4864.

Settembre C, Di Malta C, Polito VA, Arencibia MG, Vetrini F, Erdin S. Ballabio A. TFEB Links Autophagy to Lysosomal Biogenesis. Science 2011; 332(6036), 1429-1433.

Zinke I, Schütz CS, Katzenberger JD, Bauer M, Pankratz MJ. Nutrient control of gene expression in *Drosophila*: microarray analysis of starvation and sugar-dependent response. EMBO J. 2002;21(22):6162-6173.

6) In general, the paper suffers from a lack of transparency with regard to statistical methods. In particular, while some of the bar charts include all the data (e.g. Figure 1F), almost all the others do not (e.g. all the figures showing mean percentage winged aphids). Although the sample sizes are provided in the legends, standard error bars are not sufficient to capture the true data. Box plots capture both means, but also the distribution of the underlying data, and would be more appropriate for this study. The authors should also provide all their data as a Supplementary file, not just the gene expression data.

Thanks for the comments. Box plots have now been included in all related figures in the main text and supplementary files (Figure 1A-C, Figure 1F, Figure 2B-I, Figure 3H-N, Figure 4A-H, Figure 1—figure supplement 1, Figure 3—figure supplement 1).

Reviewer #2 (Recommendations for the authors):In the introduction, I think a more thorough discussion of the Target of Rapamycin pathway would help. Currently, the connection between TOR and REPTOR is not clear.

We have added some information as you suggested (see Line 78-92).

In the results text, it would be useful to provide a bit more information about why you reduced amino acids in Figure 4G. I understand that you expect reducing the amino acid concentration of the diet to reduce TOR signalling, but this isn't explicitly stated when you introduce the experiment.

Deprivation of dietary amino acids could suppress aphid TOR signaling, by which autophagy was activated and winged proportion was decreased. Modifications have been made in the text as suggested (see Line 217-219).

Finally, to show that your manipulations of TOR signalling occur through REPTOR2, and not through a parallel mechanism, it would useful to (1) examine REPTOR2 expression in the dsTOR animals and (2) perform the double knockdowns (dsREPTOR2 and dsTOR).

As suggested, we examined REPTOR2 expression in dsGFP and dsTOR aphids, and found that suppression of TOR increased the REPTOR2 transcripts in the wing disc and whole body by 1.97-fold and 1.75-fold, respectively (Figure 4K and 4L). In addition, aphids with both TOR and REPTOR2 silenced had lower wing disc autophagy and higher winged proportion when compared with dsTOR aphids (Figure 4I and 4J).